# A Systematic Pan-Cancer Analysis of MEIS1 in Human Tumors as Prognostic Biomarker and Immunotherapy Target

**DOI:** 10.3390/jcm12041646

**Published:** 2023-02-18

**Authors:** Han Li, Ying Tang, Lichun Hua, Zemin Wang, Guoping Du, Shuai Wang, Shifeng Lu, Wei Li

**Affiliations:** 1Key Laboratory of Environmental Medicine Engineering, Department of Epidemiology and Health Statistics, School of Public Health, Southeast University, Nanjing 210009, China; 2Department of Ultrasound Diagnostic, Children’s Hospital of Nanjing Medical University, Nanjing 210008, China; 3Department of General Practice, Southeast University Hospital, Nanjing 210018, China; 4Department of Hematology and Oncology, Children’s Hospital of Nanjing Medical University, Nanjing 210008, China; 5Department of Clinical Research, Children’s Hospital of Nanjing Medical University, Nanjing 210008, China

**Keywords:** MEIS1, pan-cancer analysis, tumor, biomarkers, immunotherapy, prognosis

## Abstract

Background: We intended to explore the potential immunological functions and prognostic value of Myeloid Ecotropic Viral Integration Site 1 (MEIS1) across 33 cancer types. Methods: The data were acquired from The Cancer Genome Atlas (TCGA), Genotype-Tissue Expression (GTEx) and Gene expression omnibus (GEO) datasets. Bioinformatics was used to excavate the potential mechanisms of MEIS1 across different cancers. Results: MEIS1 was downregulated in most tumors, and it was linked to the immune infiltration level of cancer patients. MEIS1 expression was different in various immune subtypes including C2 (IFN-gamma dominant), C5 (immunologically quiet), C3 (inflammatory), C4 (lymphocyte depleted), C6 (TGF-b dominant) and C1 (wound healing) in various cancers. MEIS1 expression was correlated with Macrophages_M2, CD8+T cells, Macrophages_M1, Macrophages_M0 and neutrophils in many cancers. MEIS1 expression was negatively related to tumor mutational burden (TMB), microsatellite instability (MSI) and neoantigen (NEO) in several cancers. Low MEIS1 expression predicts poor overall survival (OS) in adrenocortical carcinoma (ACC), head and neck squamous cell carcinoma (HNSC), and kidney renal clear cell carcinoma (KIRC) patients, while high MEIS1 expression predicts poor OS in colon adenocarcinoma (COAD) and low grade glioma (LGG) patients. Conclusion: Our findings revealed that MEIS1 is likely to be a potential new target for immuno-oncology.

## 1. Introduction

Cancer is a disease that seriously affects human health, and it is also the most concerning disease by global research [1]. Globally, 18.1 million cases of cancer were newly diagnosed in 2020, and this number is expected to be 28.4 million in 2040; with the burden growing in almost every country, the prevention and treatment of cancer are significant public health challenges [2]. Currently, cancer treatment approaches primarily include chemotherapy, radiotherapy, surgery, targeted therapy, stem cell transplants, and immunotherapy [3,4]. Recently, immunotherapy has shown positive outcomes in various cancer types previously limited due to lack of known intervention strategies [5]. Immunotherapy is a form of cancer therapy that helps human to fight cancer by activating the immune system, which is assuredly one of the greatest innovation in cancer treatment in the last decade [6]. Biomarkers are important for predicting the outcomes in response to immunotherapy, numerous candidate biomarkers have been used; however, responses are only typically present in a small number of patients [7]. Novel immunotherapy targets can be screened by conducting pan-cancer analysis of genes in human tumors [8], including those that characterize the tumor microenvironment and targeted signaling pathways [9].

MEIS proteins, which belong to the three amino acid loop extension (TALE) class of transcription factor family, whose members include MEIS1 (NC_000002.12, Appendix A), MEIS2, and MEIS3 [10]. MEIS1 was first described in a leukemia mouse model. According to the single cell type module of Human Protein Atlas (http://www.proteinatlas.org/, accessed on 12 January 2023), MEIS1 expression is specificity to oligodendrocytes, rod photoreceptor cells, and endometrial stromal cells (Appendix A). Meanwhile, in different blood and immune cells, the expression of MEIS1 was higher in total peripheral blood mononuclear cells (PBMC), natural killer (NK)-cells, and plasmacytoid dendritic cells (DC) (Appendix A).

The expression of MEIS1 was influenced by cell types, age, the environment humans stay in, their pathological state, and the metabolism features of cancer cells [10,11]. MEIS1 was engaged in numerous cellular processes including chromatin remodeling, cell cycle regulation, apoptosis, and transcription regulation of self-renewal genes [12]. Research has shown that MEIS1 was strongly related to HOX genes and their cofactors to exert its regulatory effects on numerous signaling pathways [13]. Moreover, MEIS1 was a directly repressed target of MYC, and via effects on *HOXB13*, links MYC activity to androgen receptor activity to mediate cancer development [14]. MEIS1 can promote leukemogenesis and supports leukemic cell homing and engraftment by inducing synaptotagmin-like 1 (SYTL1) [15]. The impaired expression of MEIS1 was highly correlated with the poor prognosis of colorectal cancer patients [16]. In addition, MEIS1 was identified to reduce major histocompatibility complex class II (MHCII)expression in acute myeloid leukemia (AML) cells [17]), MEIS1 overexpression improved survival in patients with AML [18].

As a transcription factor, MEIS1 drives cell growth [19], dysregulated MEIS1 expression contributes to tumorigenesis in multiple tumor types [12,20,21,22,23]. Due to its role in cancer cell proliferation, MEIS1 can become a molecular biomarker for cancer diagnosis and even a target for cancer therapy [24]. However, the expression levels of MEIS1 in various tumors are different [10,24]. It is a negative regulator in non-small-cell lung cancer (NSCLC) [25], whereas it serves as a positive regulator in esophageal squamous cell carcinoma (ESCC) [26], malignant peripheral nerve sheath tumor (MPNST) [19] and Ewing sarcoma [27]. Even different studies found inconsistent expression of MEIS1 in prostate cancer [28,29]. The function of MEIS1 in cancer needs reassessing, we speculated that the oncogenic role of MEIS1 was affected by multiple factors, and it is possible that the complex role of MEIS1 in proliferation may largely depend on the tumor microenvironment (TME) [30]; nevertheless, the function of MEIS1 in TME remains uncertain. Various online tools were used to analyze the data from TCGA, GTEx and GEO databases, which intended to explore the potential mechanisms of MEIS1 across different cancers.

## 2. Materials and Methods

### 2.1. The Analysis of MEIS1 Expression in Different Cancers

MEIS1 gene differential expression between tumor tissues and normal tissues in 33 tumor types was explored in Gene_DE module of Tumor Immune Estimation Resource 2.0 (TIMER 2.0) [31]. Concerning many tumors without control data in TIMER 2.0, the Gene Expression Profile module of Gene Expression Profiling Interactive Analysis version 2 (GEPIA2) was used to match TCGA with GTEx data [32]. The differential expression analysis method was ANOVA and log2 (TPM + 1) for log-scale was used (we set |Log2FC| Cutoff as 1 with q-value Cutoff of 0.01) for box plots. “Stage Plots” in GEPIA2 can present MEIS1 gene expression in different stages in 26 tumor types. Log2(TPM + 1) for log-scale was used for violin plots.

The University of Alabama at Birmingham Cancer (UALCAN) data analysis Portal was used to analyze MEIS1 protein expression, phosphoprotein level and promoter methylation [33]. Breast cancer, clear cell renal cell carcinoma (ccRCC), colon cancer, lung adenocarcinoma (LUAD), ovarian cancer, and uterine corpus endometrial carcinoma (UCEC) were available for total-protein, phosphorylation and promoter methylation level analyses.

### 2.2. Association of MEIS1 with Survival in Different Tumors

In GEPIA2, the “Survival Analysis” module was used to reveal OS and disease-free survival (DFS) curves using the Kaplan–Meier method for the high and low MEIS1 expression groups in different cancer types. High and low expression groups were classified according to a 50% (median) cutoff of MEIS1 expression, and the hypothesis test method was the log-rank test.

### 2.3. Gene Alteration and Immune Infiltration Analysis of MEIS1

cBioPortal (https://www.cbioportal.org/, accessed on 28 September 2022) was equipped with the function to analyze gene alteration frequency, alteration types (mutation, amplification, multiple alterations), and mutation sites. The expression of MEIS1 in 22 tumor-infiltrating immune cells (TIICs) was evaluated by CIBERSORT on the SangerBox website (http://sangerbox.com/Tool, accessed on 28 September 2022). Meanwhile, the relationship between the expression of MEIS1 and immune checkpoint genes (ICPGs), TME biomarkers including TMB, MSI, and NEO were also explored. Furthermore, the relevance between MEIS1 expression and immune subtypes of 30 tumor types and molecular subtypes of 17 tumor types were analyzed utilizing the TISIDB database (http://cis.hku.hk/TISIDB/index.php, accessed on 28 September 2022).

### 2.4. Gene Enrichment Analysis of MEIS1

Firstly, the top-50 proteins binding with MEISI1 were obtained in STRING (https://string-db.org/, accessed on 7 October 2022) by setting “the minimum required interaction score”, “the meaning of network edges”, “active interaction sources” and “the max number of interactors to show” to “Low confidence”, “Evidence”, “Experiments” and “No more than 50 interactors”, respectively. The top-50 MEIS1-binding genes network was also presented.

The “Similar Gene Detection” module of GEPIA2 was used to detect 100 similar genes to MEIS1. In addition, the first five genes correlated with MEIS1 were selected according to Pearson’s correlation coefficients using the “Correlation Analysis” module of GEPIA2.

VENNY2.1 (https://bioinfogp.cnb.csic.es/tools/venny/index.html, accessed on 7 October 2022) was used to compare interacted genes with similar genes. GO and KEGG analysis was performed on 147 genes in two sets. “org.Hs.eg.db” package was applied to ID transform and the “clusterProfiler” package was applied to enrichment analysis. Data visualization was achieved through the “ggplot2” packages. This analysis was realized by R software [R-4.1.0, 64-bit, Vienna, Austria].

### 2.5. Ethical Approval

Our study was conducted in accordance with the principles stated in the Declaration of Helsinki, the secondary analysis of existing data of public use data sets does not require informed consent and ethical approval.

## 3. Results

### 3.1. Differential Expression of MEIS1 in Cancers

The result of MEIS1 expression in different cancers by the TIMER2.0 webserver was shown in Figure 1a, MEIS1 was down-regulated (the gene expression in the tumor group was lower than that in the control group) in bladder urothelial carcinoma (BLCA), breast invasive carcinoma (BRCA), COAD, HNSC, kidney renal papillary cell carcinoma (KIRP), LUAD, lung squamous cell carcinoma (LUSC), prostate adenocarcinoma (PRAD), rectum adenocarcinoma (READ), thyroid carcinoma (THCA) and uterine corpus endometrial carcinoma (UCEC) (*p* < 0.001), pheochromocytoma and paraganglioma (PCPG) (*p* < 0.01), stomach adenocarcinoma (STAD), cervical squamous cell carcinoma and endocervical adenocarcinoma (CESC) (*p* < 0.05). MEIS1 expression in tumor tissues of cholangiocarcinoma (CHOL), Liver hepatocellular carcinoma (LIHC) (*p* < 0.001) and glioblastoma multiforme (GBM) (*p* < 0.05) was higher than normal tissues.

For those tumors without control tissues in TIMER 2.0 datasets, GEPIA2 was used to match these tumor tissues with the control tissues in the GTEx database to obtain boxplots of expression differences. As shown in Figure 1b (*p* < 0.05), MEIS1 was differentially expressed in ACC, acute myeloid leukemia (LAML), skin cutaneous melanoma (SKCM), testicular germ cell tumors (TGCT), thymoma (THYM) and uterine carcinosarcoma (UCS), which was down-regulated in ACC, SKCM, TGCT and UCS, and up-regulated in LAML and THYM. Furthermore, the statistically significant results were verified through GEO database. These results were presented in Appendix A, which were consistent with the results of MEIS1 differential expression analysis using the TCGA and GTEx database. MEIS1 protein expression in six common tumors was executed utilizing the CPTAC dataset. Figure 1c showed the total MEIS1 protein expression in breast cancer, colon cancer, LUAD, UCEC and ccRCC was higher than that in normal tissues (*p* < 0.001). Figure 1d revealed the correlation between pathological stages and MEIS1 expression levels, including ACC, COAD, KIRC, KIRP, and LIHC (*p* < 0.05). The correlation between pathological stages and MEIS1 expression levels in other cancer types with no statistical significance were shown in Appendix A.

### 3.2. Prognostic Index of MEIS1 in Different Cancers

As shown in Figure 2a, GEPIA2 survival analysis showed that low MEIS1 expression predicts poor OS of ACC, HNSC and KIRC patients (*p* < 0.01), while high MEIS1 expression predicts poor OS of COAD patients (*p* < 0.05) and LGG patients (*p* < 0.001). As shown in Figure 2b, high MEIS1 expression predicts poor DFS of COAD and KIRP patients (*p* < 0.01), and LGG patients (*p* < 0.001), while low MEIS1 expression predicts poor DFS of HNSC patients (*p* < 0.05).

### 3.3. Alteration of MEIS1 Gene Analysis Data

According to Figure 3a, in NSCLC, the frequency of gene change was the highest (3.51%) and mutation is the main type of gene change. Mutation is the main type of genetic change in ESCC, melanoma, endometrial carcinoma, esophagogastric adenocarcinoma, colorectal cancer, cervical squamous cell carcinoma, adrenal cortical cancer and hepatocellular carcinoma as well. Likewise, amplification became the primary type of MEIS1 gene alteration in ovarian epithelial tumor, bladder urothelial carcinoma, mature B-cell neoplasms and prostate adenocarcinoma. Furthermore, Figure 3b contains more details, such as mutation types, sites and the number of cases with “Missense” as the main type of mutation. R102Afs*20 mutation can be observed in 2 COAD patients while R102Pfs*18 can be detected in a UCEC patient. We did not find any hotspot mutation in the MEIS1 gene.

### 3.4. Phosphorylation and Promoter Methylation Expression of MEIS1 across Different Cancers

Changes in phosphorylation pathway are closely related to cancer. Based on the CPTAC dataset, breast cancer, ovarian cancer, colon cancer, ccRCC, UCEC, LUAD and pediatric brain cancer were included to explore phosphorylation level between their tumor tissues and normal tissues. Ultimately, the box plots of four types of cancer are available in Figure 4a. In ovarian cancer, MEIS1 phosphoprotein level (S194, S196 and T202) between tumor and normal tissues has no statistical differences, while the phosphorylation level of S196 in LUAD (*p* < 0.001), ccRCC (*p* < 0.001) and UCEC (*p* < 0.001) were higher in normal tissues compared to tumor tissues. More experiments evidence is needed to identify the function of MEIS1 phosphorylation at the S196 site in tumorigenesis. In addition, it is found that promoter methylation level of MEIS1 in BLCA, HNSC, KIRC, KIRP, PRAD and UCEC were lower in primary tumors compared to normal tissues (*p* < 0.05) (Figure 4b).

### 3.5. MEIS1 Expression in Immune and Molecular Subtypes across Different Cancers

As shown in Figure 5a, the expressions of MEIS1 were different in various immune subtypes including C1-C6 in ACC, BLCA, BRCA, CHOL, COAD, KIRC, LGG, LUAD, LUSC, pancreatic adenocarcinoma (PAAD), PRAD, sarcoma (SARC), STAD, TGCT (all *p* < 0.05). In addition, MEIS1 was diversely expressed in multiple molecular subtypes of ACC, BRCA, esophageal carcinoma (ESCA), GBM, HNSC, KIRP, LGG, LIHC, LUSC, ovarian serous cystadenocarcinoma (OV), PCPG, PRAD, STAD and UCEC (all *p* < 0.05) (Figure 5b).

### 3.6. MEIS1 Expression Is Related to ICPGs Expression in Different Cancers

Subsequently, the correlation between MEIS1 expression and 60 ICPGs expression was explored (Figure 6). In most cancers, such as lymphoid neoplasm diffuse large B-cell lymphoma (DLBC), PRAD, READ, COAD, OV, KIRC and LIHC, MEIS1 expression was positively related to the expression of most ICPGs. It means that the patients with high expression of MEIS1 will show better immunotherapy effects by using immune checkpoint inhibitors (ICIs). However, in TCGT and SARC, MEIS1 expression was negatively correlated with most ICPGs expression. It showed that patients with high MEIS1 expression will have a poor prognosis when targeting these ICPGs. Therefore, it can be proved that MEIS1 is equipped with strong potential in immunotherapy.

### 3.7. The Correlation between MEIS1 Expression and Immune Cell Infiltration and TME in Different Cancers

According to results in Figure 7, MEIS1 expression was closely correlated with most immune cells in various cancers. The expression of MEIS1 was correlated with macrophages_M2 in 20 cancer types, CD8+T cells in 19 cancer types, Macrophages_M1 in 15 cancer types, macrophages_M0 in 13 cancer types and neutrophils in 10 cancer types. For evaluating anti-tumor immunity, as shown in Figure 8, the relationships between MEIS1 and TMB, MSI and NEO were explored in all cancers. MEIS1 expression was negatively associated with TMB in ACC, STAD, STES, SARC, BLCA and KIRC. As for MSI, MEIS1 expression was negatively correlated with MSI in UCS, UCEC, SARC, STAD, STES, GBM, LGG, KIPAN, PRAD and HNSC, while it was positively associated with MSI in TGCT. In addition, MEIS1 expression had a negative correlation with NEO in SARC, UCEC, KIRP, KIPAN, KIRC and LUAD.

### 3.8. MEIS1-Related Genes Are Correlated to Cell Proliferation and Differentiation

Based on STRING, 50 MEIS1-binding proteins supported by experiments were obtained. The protein–protein interaction networks are presented in Figure 9a. After that, 100 similar genes were acquired from correlation analysis in GEPIA2, 50 interacted genes and 100 similar genes have 3 overlap genes in Figure 9b. Then, the 147 genes were integrated to perform GO and KEGG analysis presented in Figure 9c. KEGG analysis indicates that “Transcriptional misregulation in cancer” might play a vital role in the effect of MEIS1 on cancers. Furthermore, by GO analysis, at the biological process (BP) level, most of the genes are involved in cell differentiation and cell fate determination. At the molecular function (MF) level, most of the genes took part in regulating DNA-binding transcription activator and repressor activity, RNA polymerase II-specific, enhancer sequence-specific and activating transcription factor binding. At last, the cellular component (CC) also proved that these genes were related to transcription.

## 4. Discussion

Cancer is a leading cause of death in each country, and the number of newly confirmed patients continues to increase [2]. Although current cancer therapy options exhibit some clinical success, a large number of cancer patients still have poor prognoses because of drug resistance, adverse effects, and other issues [34]. Consequently, there is a need to identify new therapeutic targets and sensitive biomarkers for cancer diagnosis and treatment. The pan-cancer analysis can deliver deep perception for the design of cancer prevention and precision treatment strategies by revealing the similarities and differences between different cancers [34]. In our study, we comprehensively performed MEIS1 expression and its correlation with prognostic and immunotherapy value in pan-cancer via different databases. The 33 cancer types were studied to identify MEIS1 expression in this study and it finally suggests that MEIS1 expression is different between tumor tissues and normal tissues in 23 cancer types. Among these 23 cancer types, MEIS1 expression is down-regulated in 18 cancer types, on the contrary, the other 5 types were up-regulated.

Another major finding is that MEIS1 expression correlates with survival in cancer patients, which indicated that the abnormal expression of MEIS1 can be responsible for the poor prognosis of tumors. For the value of MEIS1 on the prognosis of cancer patients, it could be determined by the activity of the MEIS1 protein which depends on the environment in which the cell exists. A similar finding was also demonstrated by Schulte et al. and Meng et al. by performing studies on MEIS1 [30,35]. MEIS1 is a hub gene in the differentially expressed gene interaction network for lung adenocarcinoma (LUAD) and participated in the occurrence and prognosis of LUAD [23]. MEIS1 overexpression was inversely correlated with relapse and overall survival in children with acute leukemias [36]. Down-regulated expression of MEIS1 was detected in colorectal cancer and predicted poor survival of colorectal cancer patients [16]. It is believed that abnormal DNA methylation plays a critical role in cancer development and progression [37]. Tumors and benign tissues exhibit different methylation patterns [38,39]. Our study found that the promoter methylation levels of MEIS1 in BLCA, HNSC, KIRC, KIRP, PRAD, and UCEC were lower in primary tumors compared to normal tissues. In conclusion, MEIS1 may be able to regulate some tumors at an epigenetic level, but more studies are needed to clarify the deeper mechanisms.

Immune infiltration is highly associated with the prognosis of tumor patients [40]. Six types of immune infiltration had been explored in cancer patients, maybe promoting or inhibiting tumor cell growth [41]. Among them, C5 is positively correlated to MEIS1. A high level of C5 can impair the curative effect of immune checkpoint inhibitors, as reported by Lehrer and his colleagues [42]. Our findings showed that MEIS1 expression was strongly related to most immune cells in various types of cancer. Moreover, MEIS1 expression is related to some immune subtypes and molecular subtypes in different types of cancer. We indicated that MEIS1 can play a role in the growth and progression of cancer and have a close correlation with immune regulation. The TMB, MSI, and NEO which characterize anti-tumor immunity were negatively related to MEIS1 expression in some cancers. Therefore, the immunotherapeutic effect of cancer patients can be predicted according to MEIS1 expression. It can contribute to guide immunotherapy more accurately. The MEIS1 expression was negatively related to TMB, MSI, and NEO in SARC. Thus, there is an inference that patients suffering from SARC with low MEIS1 expression might predicted better survival after immunotherapy. In most cancers, MEIS1 expression was positively related to most ICPGs expression. There is bold speculation that using immune checkpoint inhibitors along with interfering in MEIS expression will have an effective impact on the patient [30].

In gene enrichment analysis, HOXD4, HOXB4, and PBX2 were co-expressed with MEIS1. HOX were transcription factors that serve as an essential regulator for cell fate determination, stem cell functions, and gastrointestinal development. And they need TALE family proteins such as MEIS and PBX to improve their transcriptional efficiencies [43]. MEIS and PBX were correlated with cell growth, differentiation and apoptosis [44,45]. MEIS1 accompanied by HOX protein can form a complex to recruit transcriptional corepressors or coactivators [13]. Meanwhile, the activation of MEIS1 always couples with the activation of HOXA7 or HOXA9 [46]. Previous papers have summarized the impact of HOX genes on tumors and concluded that HOX genes take part in the development of different tumors [47]. There was research which found that MEIS1 and PBX2 overexpressed in nephroblastomas [48]. So, there may be interference with HOX genes and PBX genes during the role of MEIS1 in cancers. A further mechanism of MEIS1 for different tumors is needed, to explore in the future.

According to KEGG analysis in this study, “Transcriptional misregulation in cancer” might play a vital role in the effect of MEIS1 on cancers. As an important transcription factor, if MEIS1 is over-expressed or low-expressed, transcriptional misregulation will emerge and then play a role in cancer development [49]. Based on GO analysis, MEIS1 may take part in tumor development by regulating cell differentiation and “DNA-binding transcription activator and repressor activity”, “RNA polymerase II-specific”, “enhancer sequence-specific” and “activating transcription factor binding”. Several researches have shown that MEIS1 serves as a tumor suppressor in ccRCC [50], prostate [29], NSCLC [25], gastric [51], and colorectal cancers [52] by promoting cell differentiation and inhibiting epithelial cell proliferation.

## 5. Conclusions

To sum up, our findings offer a comprehensive understanding of the functions of MEIS1 in prognostic and immunotherapy in different types of cancer. MEIS1 has the potential as a cancer immunotherapy target and is worthy of more attention. Moreover, our results may provide valuable theoretical guidance for further research on the mechanism of MEIS1 in vivo and in vitro.

## 6. Limitations

This research had some restrictions. Initially, we only performed a series of bioinformatics analysis of MEIS1 in multiple databases, the mechanisms of MEIS1 in different cancer types were not verified. Secondly, the small samples for some tumor types may cause inaccurate results or bias.

## Figures and Tables

**Figure 1 jcm-12-01646-f001:**
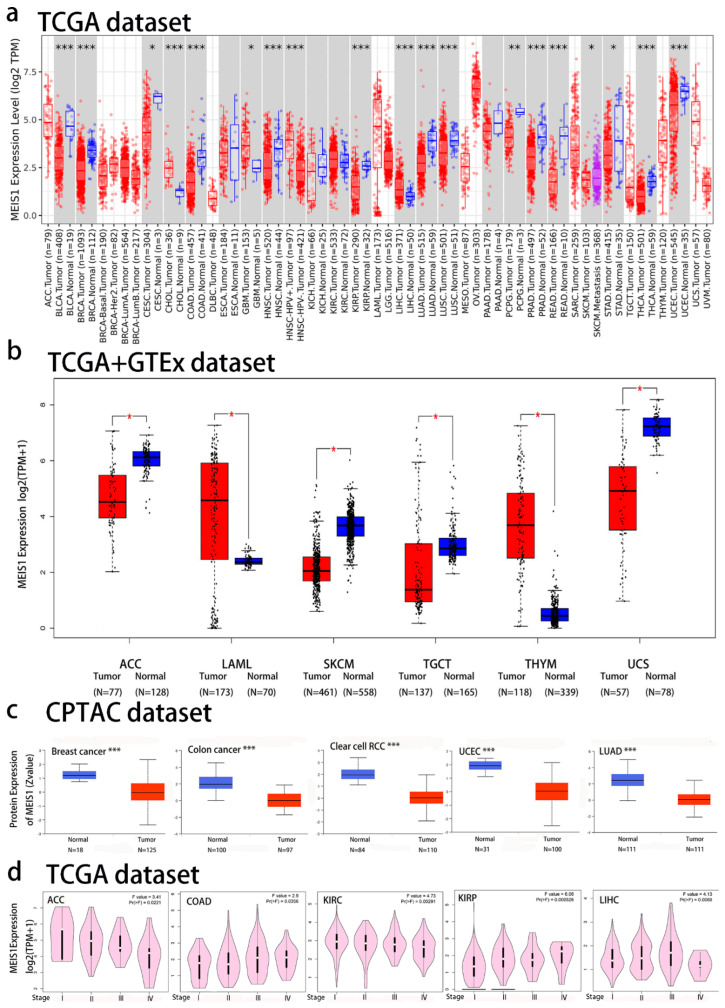
MEIS1gene and protein expression in 33 tumors and pathological stages of tumors. (**a**) MEIS1 gene expression level in 33 tumor tissues and normal tissues, even specific subtypes of tumors through TIMER2 (Blue represents normal tissues, red represents tumor tissues and purple represents metastatic tumors). *** *p* < 0.001; ** *p* < 0.01; * *p* < 0.05. (**b**) MEIS1 gene expression in tumors without normal tissues in TCGA, including ACC, LAML, SKCM, TGCT, THYM and UCS. * *p* < 0.05. (**c**) Total protein expression level of MEIS1 in tumor tissues and normal tissues of Breast cancer, Colon cancer, ccRCC, UCEC and LUAD. *** *p* < 0.001. (**d**) MEIS1 expression level in different pathological stages of ACC, COAD, KIRC, KIRP and LIHC.

**Figure 2 jcm-12-01646-f002:**
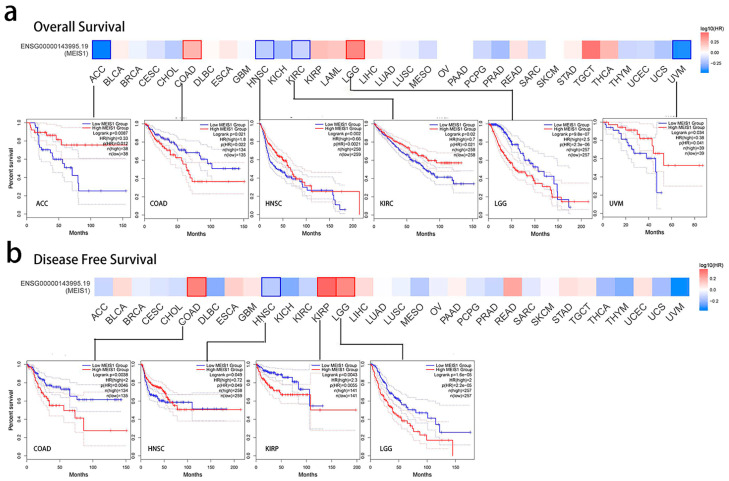
Correlation between MEIS1 expression and survival of patients of cancers in TCGA. The survival plots and Kaplan-Meier curves with positive outcomes were given. (**a**) Correlation between MEIS1 expression and OS of ACC, COAD, HNSC, KIRC, LGG and UVM patients. (**b**) Correlation between MEIS1 expression and DFS of COAD, HNSC, KIRP and LGG patients.

**Figure 3 jcm-12-01646-f003:**
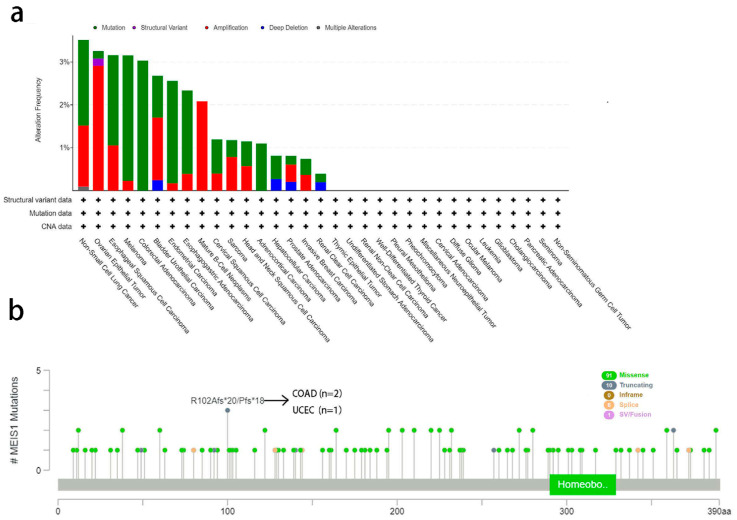
MEIS1 gene alteration frequency, types, sites and cases. (**a**) MEIS1 alteration types and frequency. (**b**) MEIS1 mutation types, sites and cases. “+” represents that data are available.

**Figure 4 jcm-12-01646-f004:**
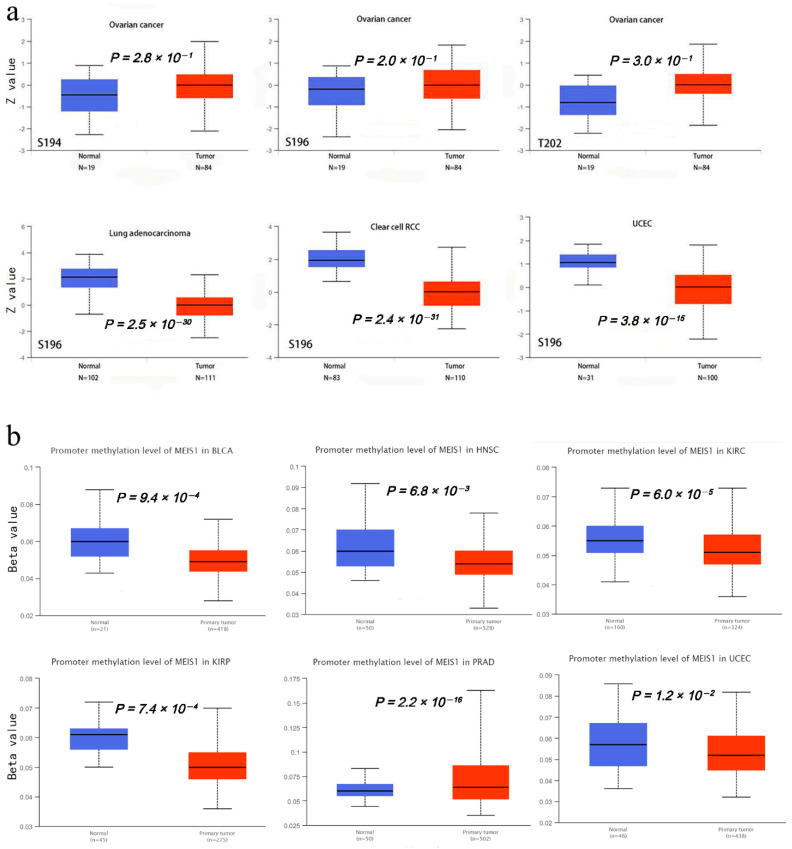
Phosphorylation and promoter methylation analyses of MEIS1 protein in different tumors. (**a**) Phosphorylation level of MEIS1 between their tumor tissues and normal tissues. (**b**) Promoter methylation level of MEIS1 between their tumor tissues and normal tissues.

**Figure 5 jcm-12-01646-f005:**
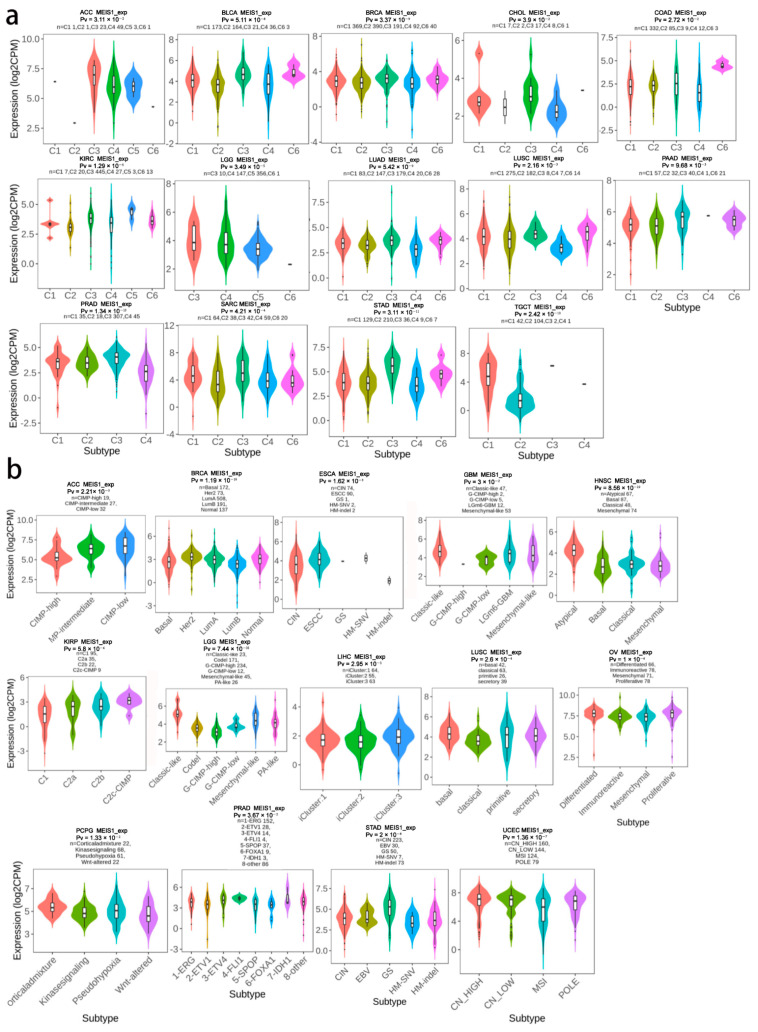
MEIS1 expression in immune and molecular subtypes of cancers (**a**) The relationship between MEIS1 expression and immune subtypes of cancers. (**b**) The relationship between MEIS1 expression and molecular subtypes of cancers.

**Figure 6 jcm-12-01646-f006:**
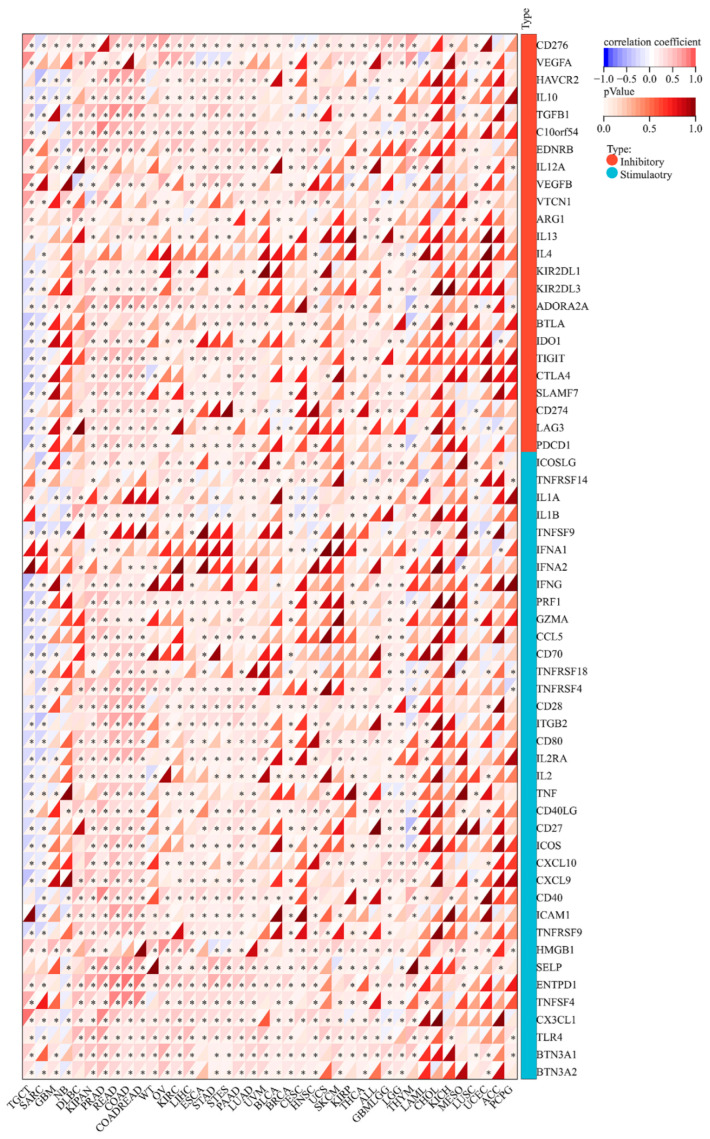
The correlation matrix between MEIS1 expression and ICPG expressions. * *p* < 0.05.

**Figure 7 jcm-12-01646-f007:**
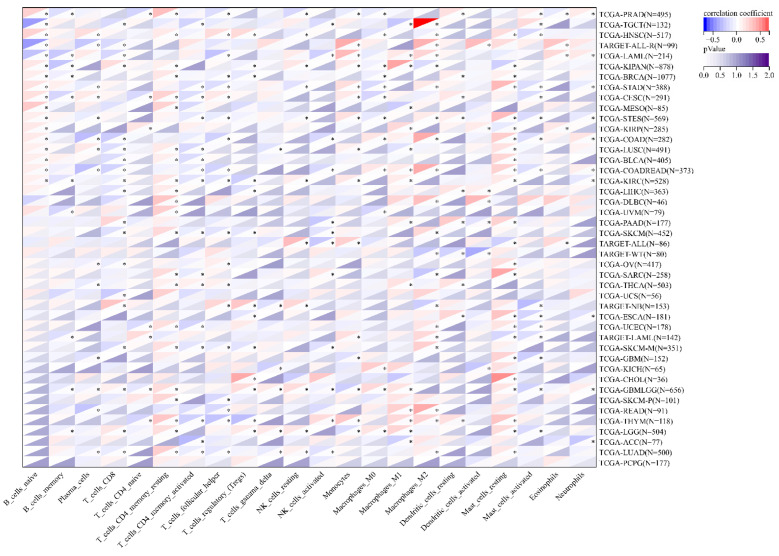
The correlation matrix between MEIS1 expression and immune cell content. * *p* < 0.05.

**Figure 8 jcm-12-01646-f008:**
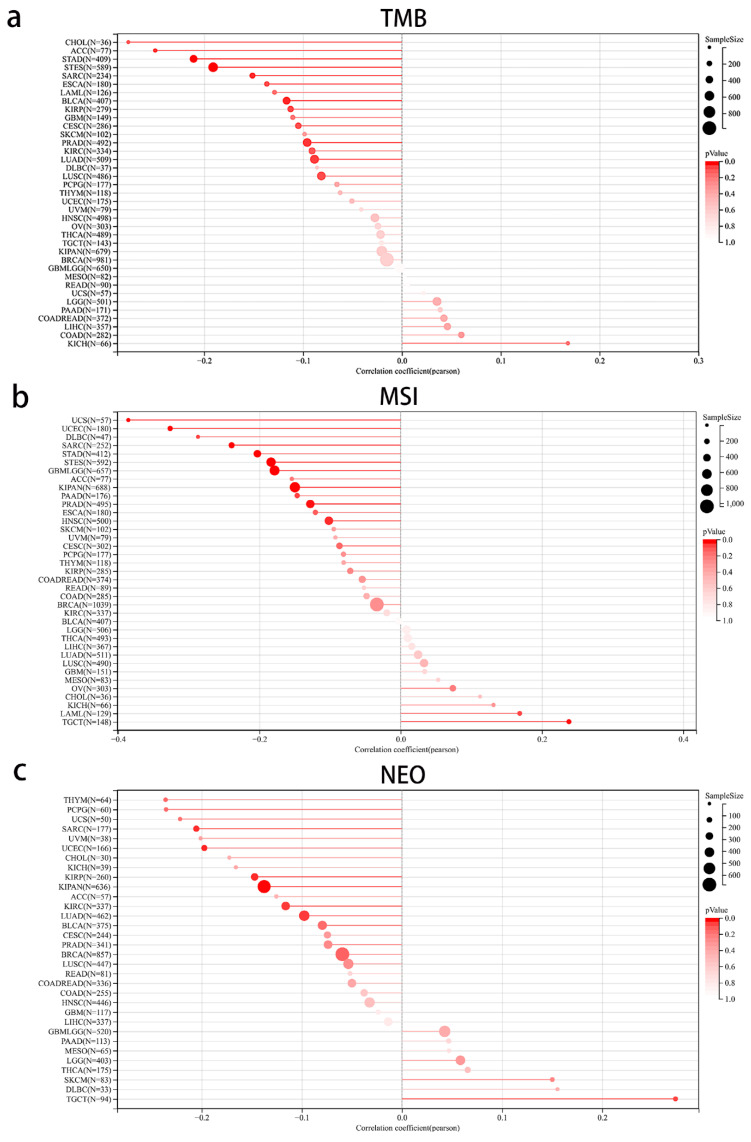
The relationship between MEIS1 expression and anti-immunity indicators (**a**) The relationship between MEIS1 expression and TMB (**b**) The relationship between MEIS1 expression and MSI. (**c**) The relationship between MEIS1 expression and NEO.

**Figure 9 jcm-12-01646-f009:**
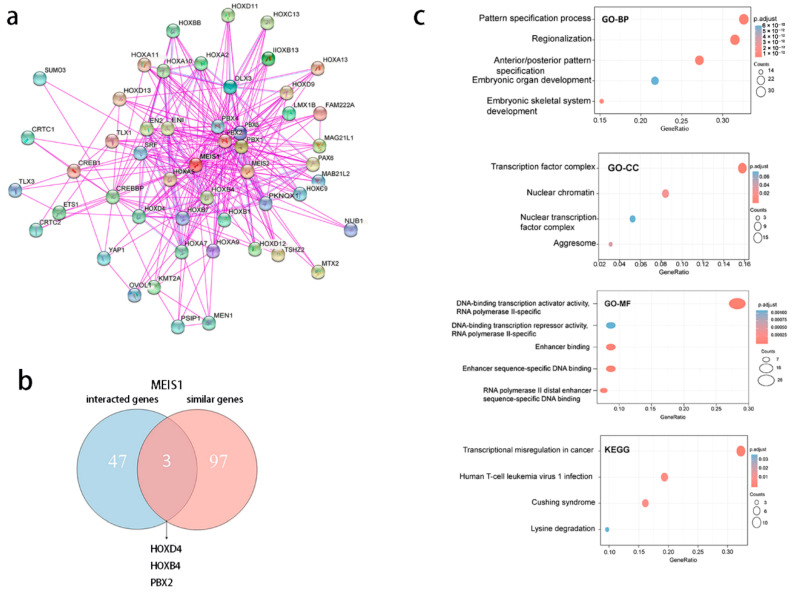
MEIS1-related genes’ enrichment analysis. (**a**) 50 MEIS1-binding proteins’ interaction network derived from STRING. (**b**) Venn diagram showing the intersection between MEIS1-interacted genes and MEISI-correlated genes. (**c**) Bubble diagram for GO and KEGG analysis of MEIS1-related genes.

## Data Availability

The datasets presented in this study can be found in online repositories. The names of the repository/repositories and accession number(s) can be found in the article.

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
