# Peer review of "A Systematic Pan-Cancer Analysis of MEIS1 in Human Tumors as Prognostic Biomarker and Immunotherapy Target"

_jcm, 2023, doi:10.3390/jcm12041646_

Round 1
Reviewer 1 Report (Previous Reviewer 2)
I saw some minor English writing issues.
Reviewer 2 Report (Previous Reviewer 1)
None
This manuscript is a resubmission of an earlier submission. The following is a list of the peer review reports and author responses from that submission.
Round 1
Reviewer 1 Report
Please see the file attached.

Reviewer 2 Report
It is interesting that I haven’t seen any information related to leukemia, since it is well-known that Meis1 is over-expressed in leukemia. This could somehow give an idea about the over-expression of Meis1 in other tumors.
Lane 84 “It is a negative regulator in non- small cell lung cancer (NSCLC)(Li et al., 2014) and prostate cancer(VanOpstall et al., 2020), whereas it serves as a positive regulator in esophageal squamous cell carcinoma(ESCC)(Rad et al., 2016), malignant peripheral nerve sheath tumor (MPNST) (Patel et al., 2016) and Ewing sarcoma(Lin et al., 2019).”
Meis1 function in prostate cancer is not quite clear, there is also a study stating that Meis1 promotes cancer progression in prostate cancer (Johng et.al, 2019).
The explanation of some abbreviations is not included throughout the entire text. For example, in Figure 1, what is MESO?
It is more appropriate to include the 23 significant cancer types found in Figure 1a at the beginning of the analysis and the remaining 10 cancer types to be at the end of the list or to exclude them from the analysis as no result can be reached.
Lane 165 “Protein expression analysis of six common tumors was performed using the CPTAC dataset. Fig.1c showed the expression of total MEIS1 protein in Breast cancer, Colon cancer, Lung adenocarcinoma, UCEC and Clear cell RCC was higher than that in normal tissues (P<0.001)”
I don't see the sixth tumor. Besides if it is six common tumors, there should be data related to prostate, stomach, and liver cancer.
Lane 168 “Fig.1d revealed the correlation between pathological stages and MEIS1 expression levels, including ACC, COAD, KIRC (Kidney renal clear cell carcinoma), KIRP, and LIHC (P<0.05).”
Here the authors present some data, but they don’t explain why they picked these cancers. Did they perform analysis with other cancers, and these are the only significant ones?
In figure 2a, what is the association of Meis1 with the OS prognosis of KICH, PRAD, and TGCT?
In figure 2b, what is the association of Meis1 with the DFS prognosis of UHM and THYM?
In figure 5b some written parts of the figure are missing.
References
Johng, D., Torga, G., Ewing, C. M., Jin, K., Norris, J. D., McDonnell, D. P., & Isaacs, W. B. (2019). HOXB13 interaction with MEIS1 modifies proliferation and gene expression in prostate cancer. The Prostate, 79(4), 414-424.